# Constructing the CityGML ADE for the Multi-Source Data Integration of Urban Flooding

**Jie Shen** [1,2,3]**, Jingyi Zhou** [1,]*****, **Jiemin Zhou** [1]**, Lukas Herman** [4] **and Tomas Reznik** [4]

[1]   School of Geography, Nanjing Normal University, Nanjing 210023, China; shenjie@njnu.edu.cn (J.S.);
    161302112@stu.njnu.edu.cn (J.Z.)

[2]   Key Laboratory of Virtual Geographic Environment (Nanjing Normal University), Ministry of Education,
    Nanjing 210023, China

[3]   Jiangsu Center for Collaborative Innovation in Geographical Information Resource Development and
    Application, Nanjing 210023, China

[4]   Department of Geography, Faculty of Science, Masaryk University, Kotlarska 2, 611 37 Brno, Czech Republic;
    herman.lu@mail.muni.cz (L.H.); tomas.reznik@sci.muni.cz (T.R.)

*****   Correspondence: 191302062@stu.njnu.edu.cn

**Abstract:** Urban flooding, as one of the most serious natural disasters, has caused considerable personal injury and property damage throughout the world. To better cope with the problem of waterlogging, the experts have developed many waterlogging models that can accurately simulate the process of pipe network drainage and water accumulation. The study of urban waterlogging involves many data types. These data come from the departments of hydrology, meteorology, planning, surveying, and mapping, etc. The incoordination of space–time scale and format standard has brought huge obstacles to the study of urban waterlogging. This is not conducive to interpretation, transmission, and visualization in today's network environment. In this paper, the entities and attributes related to waterlogging are defined. Based on the five modules of urban drainage network, sub basin, dynamic water body, time series, and meteorological data, the corresponding UML (Unified Modeling Language) model is designed and constructed. On this basis, the urban waterlogging application domain extension model city waterlogging application domain extension (CTWLADE) is established. According to the characteristics of different types of data, two different methods based on FME object and citygml4j are proposed to realize the corresponding data integration, and KML (Keyhole Markup Language) /glTF data organization form and the corresponding sharing method are proposed to solve the problem that the CTWLADE model data cannot be visualized directly on the web and cannot interact in three-dimensional format. To evaluate the CTWLADE, a prototype system was implemented, which can convert waterlogging-related multi-source data in extensible markup language (XML) files conform. The current CTWLADE can map the data required and provided by the hydraulic software tool storm water management model (SWMM) and is ready to be integrated into a Web 3D Service to provide the data for 3D dynamic visualization in interactive scenes.

**Keywords:** CityGML application domain extension; waterlogging; data integration; 3D dynamic visualization

## 1. Introduction

Urban flooding refers to the phenomenon of water accumulation in cities due to heavy precipitation or continuous precipitation exceeding urban drainage capacity [1]. In the background of global climate change and rapid urbanization, the intensity and frequency of urban natural disasters and associated losses are increasing [2–4]. As one of the most serious natural disasters, urban flooding has caused considerable personal injury and property damage throughout the world. For example, in 2012, Beijing

and its surrounding areas suffered heavy rains and waterlogging disasters, flooding many buildings, and causing 79 deaths and economic losses of 1.64 billion dollars. In recent years, the acceleration of urbanization has led to a sharp increase in the urban impervious area; under the influence of extreme weather such as heavy rain and typhoon, urban flooding is highly prone to occur [5]. Therefore, the construction of reasonable disaster emergency management plans to reduce economic losses, social impacts, and human casualties caused by natural disasters have become the top priority of sustainable urban development [6,7]. In disaster management, an accurate focal priority analysis of how societies can adapt to these changing events can provide insight into practical solutions [8], risk assessment based on model simulation has become a research hotspot.

Urban waterlogging research involves multiple types of data. These data come from hydrology, meteorology, planning, surveying, and mapping departments. The spatial and temporal scales are incompatible with each other, and the format standards are not uniform, which brings huge obstacles to urban waterlogging research [9]. This is not conducive to interpretation, transmission, and visualization in today's network environment. Therefore, data integration and sharing for urban waterlogging simulation and analysis is particularly important. In order to solve this problem, some scholars have proposed a series of spatial multi-source heterogeneous data integration methods. There are three main modes of spatial data integration: Direct data access mode, data format conversion mode, and data interoperation mode [10]. The data integration model based on data standards effectively solves the inherent defects of traditional Web language in describing and expressing complex geographic information, and provides a very important way for data transmission, exchange, and integration of massive spatial data in the existing network environment Effective solution [11].

Multi-source spatial data integration has always been a hotspot in the industry. The data interoperability model based on data standards has been favored by scholars. This is because the data standards developed by extensible markup language (XML) can not only describe the spatial geometry and topological relationship of objects but also can contain rich semantic information. This makes this data integration result available for visualization and supports many analysis operations. After years of development, under the framework of smart city information, the most comprehensive standard today for the exchange of 3D city models is the CityGML standard by the Open Geospatial Consortium [12,13]. However, to use the highly extensibility of CityGML, it must integrate the urban waterlogging multi-source data with the analysis result information. It has an important research foundation to propose an integrated data model of 3D dynamic visualization.

Therefore, a proper data model, respecting the needs of integration of multisource input and output data of waterlogging simulation and analysis and the demand for 3D visualization in the front end, is needed. Although the development of data models for modeling hydrology related data has already been done, these models are currently unable to meet the above requirements. For example, ArcHydro [14], developed jointly by ESRI and CRWR, can provide a complex data model trying to consider the whole hydrologic system, but it is not based on XML (extensible markup language) or GML (geography markup language), nor is freely available; The ODM [15] (observations data model) is a relational database model of a general structure proposed by HIS (hydrologic information system) of CUAHSI (The Consortium of Universities for the Advancement of Hydrologic Science, Inc.) in the United States. It needs to observe the database structure in a unified format; WaterML [16] is an OGC (open geospatial consortium) standard for the representation of water observations data, but cannot provide the waterlogging model with input data such as underlying surface type, drainage network, etc., and does not meet the requirements of online 3D visualization.

Within this paper, we defined waterlogging-related entities and attributes, starting from the five modules of urban drainage pipe network, sub-catchment, dynamic water body, time series, and meteorological data, the corresponding UML model is designed and constructed, and then the extended model CTWLADE (city waterlogging application domain extension) for generating urban waterlogging domain is formed. According to the characteristics of different types of data, two different methods based on FME object and citygml4j are proposed to realize the corresponding data integration.

Under the two requirements of data publishing and visualization, this paper proposes KML/glTF data organization form and corresponding sharing method to solve the problem that CTWLADE model data cannot be visualized directly on the web side and cannot interact with each other in three-dimensional format. To verify the effectiveness of CTWLADE, a prototype system integrating multi-source waterlogging related data is implemented. The relevant functions of the prototype system are tested by using the data in the experimental area to verify the effectiveness and practicability of CTWLADE in the field of urban waterlogging.

## 2. Related Work

Waterlogging is accompanied by the process of urbanization, and its development process is closely related to urban climate, planning, construction, and management [17]. To accurately simulate and predict waterlogging and provide a decision-making basis for disaster prevention and urban planning, scholars try to establish a corresponding physical model to solve this issue. Early scholars established the relationship between input and output data through empirical analysis to form an empirical model to simulate urban ambiguity. This kind of model is not based on the analysis of hydrological processes and relies heavily on many historical data and fitting models, which is far from providing a reliable basis for waterlogging protection. With the deepening of the research on the mechanism of waterlogging, many scholars have summarized three hydrological processes of the waterlogging: Surface runoff yield, surface confluence, and drainage network confluence, and proposed corresponding simulation methods for each hydrological process.

(1) Calculation of surface runoff yield.

Elliott et al. [18] summarized several commonly used methods to deal with the runoff production part of urban waterlogging model and found that the calculation of the impervious area is relatively simple and unified; that is, the precipitation after deducting the loss of surface evaporation and depression interception. For the permeable area, there are many calculation methods, such as SCS method, runoff coefficient method, green Ampt method, etc. [19].

(2) Calculation of surface confluence.

For the characteristics of urban surface confluence, experts and scholars put forward two simulation methods, namely the hydrology method and the hydrodynamics method. Based on the physical mechanism of the confluence process, the hydrodynamics method solves the Saint Venant equations or its simplified form, which can describe the unsteady flow on the ground in many ways, so as to obtain more detailed rainwater surface confluence process. Because it can describe the urban surface runoff process more precisely, scholars have made a lot of useful exploration on it, and formed a variety of surface one-dimensional [20,21] or two-dimensional unsteady hydrodynamic models [22].

(3) Calculation of drainage network confluence.

It can also be divided into two methods: Hydrological method and hydrodynamic method. In the early stage of hydrology, there were many methods, such as the Muskingum method [23] and the instantaneous unit hydrograph method [24]. With the improvement of urban flood control decision-making on the accuracy of urban waterlogging simulation, more and more attention has been paid to hydrodynamic methods. At present, the most widely used are the diffusion wave equation [24] and the dynamic wave method [25], among which the successful application of the dynamic wave method in the storm water management model (SWMM) model indicates the maturity of the convergence simulation calculation technology of urban waterlogging pipe network [26].

At present, the most used waterlogging model is the distributed hydrodynamic model, such as SWMM [27–29], Mike Flood [30], InfoWorks ICM [31], Sewer GEMS [32], etc., all of which can achieve better waterlogging simulation effects. SWMM is widely used in rainfall-flood simulation and water pollution analysis in urban areas [33]. However, the use of SWMM is relatively cumbersome and limited by the operational platform, and these factors hinder the further promotion and sharing of SWMM [34].

There are still two major shortcomings of this kind of waterlogging model: (1) These models require a large amount of data to support the simulation, which is used to determine the precipitation, the type of underlying surface, and the state of the drainage network in different research areas. However, these input data often have characteristics such as multi-source, heterogeneity, multi-scale, multi-resolution, etc., which brings great difficulty to data acquisition and processing [35]. (2) The output datasets of these models are delivered in a high diversity of different formats of text-based fractionalized files containing a basic regular or irregular grid and the values at different time steps [36]. These data seem to be difficult for individuals to interpret waterlogging forecasts and local waterlogging warnings. The impact of given water heights is especially difficult to judge for non-experts. Evaluation of waterlogging warning systems indicates that visual representations are much more effective [37].

In the past few years, a standard-based, integrated three-dimensional semantic virtual city model proved to be an information carrier to meet various needs [38]. In particular, being based on open standards (e.g., on the CityGML standard proposed by OGC), virtual city models firstly reduce the effort in terms of data preparation and provision. Secondly, they offer clear data structures, ontologies, and semantics to facilitate data exchange between different domains and applications, and 3D visualization, which is essential for crisis management [39].

Due to the unique ADE (application domain extension) mechanism of CityGML, the expanding field of CityGML data model is very wide [40]. In terms of urban management, Rosser et al. used CityGML energy ADE to simulate urban residential inventory for building energy consumption simulation [41]. When CityGML application domain expansion (ADE) is used to represent the dynamic phenomenon in a three-dimensional city, the hierarchical structure of the model is extended to fully express the construction process of buildings on multiple time and space scales [42].

Although some scholars have explored the application of CityGML in flood simulation, Schulte et al. [36] proved that CityGML provides this function in three dimensions and embeds it into a semantic model by proposing the concept of CityGML application domain extension for flood-related data. Singh et al. [43] using web 3D open-source technology (such as WebGL, X3D, jQuery, and x3dom) to store 3D City database of CityGML should be integrated into the application program for quick retrieval and processing. The former focuses on semantic modeling and the latter on visual expression.

A standardized and Omni-comprehensive urban data model covering the urban waterlogging domain is still missing. Even CityGML falls partially short when it comes to the definition of specific entities and attributes (pipe sections and nodes of drainage network, catchment area, underlying surface, dynamic water body, etc.) for waterlogging-related applications.

## 3. Analysis of Urban Waterlogging

### 3.1. The Process of Urban Waterlogging

Rainfall is the source of waterlogging. During the process of heavy rainfall in the city, a small amount of rainfall evaporates in the air. Most of the rainfall is first intercepted by vegetation on the ground. When the rain reaches the ground, different runoff yield processes will occur due to the difference in urban land use type. On the permeable ground, rainwater will first penetrate the soil, and when the soil water content is saturated, the ground will produce residual water. This part of the residual water first accumulates in the depression. When the depression is filled, the remaining water will produce surface runoff. On the impervious ground, the rainwater will produce surface runoff directly or after filling the depression. After the surface runoff is generated, it will make a slope convergence to the lower slope of the terrain and be collected through the nearby urban rainwater pipe network system, such as open channels, rivers, and maintenance wells. After the underground pipe network confluence, the rainwater will eventually be discharged into the ditch or river channel. Figure 1 is a schematic diagram of the basic flow of urban production and confluence.

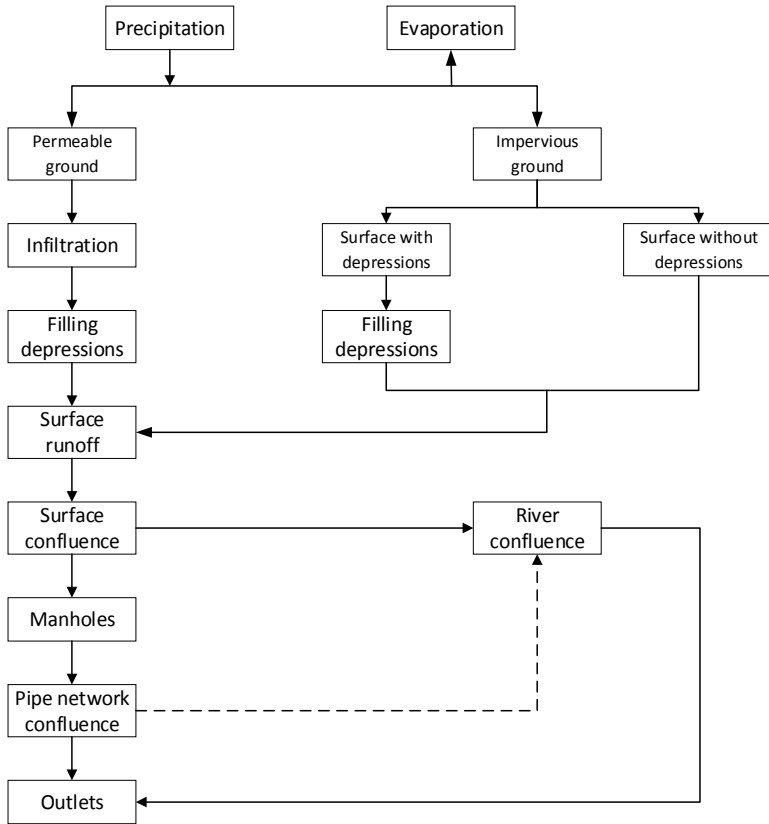

**Figure 1.** The schematic diagram of the basic flow of urban production and confluence.

Therefore, a fully functional waterlogging model should include a precipitation module, a surface runoff yield module, and a pipe network confluence module, which can support the modeling of rainfall data, land use type data, DEM (Digital Elevation Model) data, and pipe network data, and can analysis velocity of pipe network confluence, depth variation and overflow of the node, etc. The design of CTWLADE needs to consider the waterlogging related entities in the above three modules and save all kinds of data required for waterlogging simulation as attributes in the corresponding entities. At the same time, CTWLADE should also fully support time-series data to meet the needs of 3D dynamic visualization. A thorough description of each class is given in the Section 4 of the city waterlogging ADE.

*3.2. Data Sources of Urban Waterlogging*

According to the formation process of urban waterlogging and the simulation stage of waterlogging model, urban waterlogging data source can be divided into three aspects: Meteorological data, underlying surface data, and drainage pipe network data, as shown in Table 1.

Precipitation data is an important basis for regional water resources calculation. On a large scale, the limited station data cannot reflect the actual precipitation of each part of the city, so it is necessary to use an effective interpolation method to get the area rainfall. In a small scale, the observation data of meteorological stations have high accuracy.

DEM is composed of a set of plane coordinates and corresponding elevations, which is used to describe the terrain fluctuation in the study area. It is often used to extract parameters such as slope, slope direction, and flow direction, and is an important basic data for hydrological analysis.

**Table 1.** Data sources of urban waterlogging.

| Class | Type | Properties Contained |
|-------|------|----------------------|
| Meteorological data | Precipitation | Location of Meteorological Station, Time, and precipitation |
| Underlying surface data | DEM | elevation |
| | Landuse | type and area of land use |
| | Road | Shape, length, width, material, and roughness |
| Drainage pipe network data. | Drainage pipe | Pipeline No., upstream and downstream node No., elevation, length, pipe diameter and material of the beginning and end of the pipeline |
| | Drainage node | Point number, bottom elevation, and well depth |

Land use types are generally divided into the road, residential land, bare land, forest land, and grassland. The impervious areas are mainly residential areas and roads, and the permeable areas include bare land, forest land, and grassland. In the calculation of surface runoff, in addition to the influence of topography on the direction and velocity of runoff, the impact of land use type of underlying surface is more significant.

The road is the main disaster area of urban waterlogging, and also the key simulation object of urban waterlogging simulation. Therefore, high-precision vector data of a road network is still essential.

The urban drainage network is an important bearing body of urban flood control and drainage, and also an important part of confluence simulation in the urban waterlogging model. The urban drainage pipe network is mainly composed of drainage pipes and nodes.

## 4. The City Waterlogging ADE: Data Model

### 4.1. Extending CityGML Principles

CityGML is jointly released as an international data standard by OGC and ISO TC211 (International Organization for Standardization/Technical Committee 211) and realize a general information model based on an XML format for storing and exchanging virtual three-dimensional city models [44]. It defines the classification and relationship of the most relevant geographic entity objects in urban models, including not only geometry, topology, semantics, and appearance attributes [45], but also generalized and aggregated structures between different thematic, relationships between objects and spatial attributes, etc. This thematic information beyond the graphical interchange format allows virtual 3D urban models to be used for complex analytical tasks in different application areas such as simulation, urban data mining, facility management, and thematic queries.

While CityGML is designed to provide a generic geographic information model that is independent of the application, more and more applications and uses [46] require information that is not available in the CityGML data model [47]. To cope with this issue, CityGML provides two methods: (1) Storing specific information through generic objects and attributes; (2) using the application domain extension (ADE) mechanism. The former method has been used in practice, but it has serious drawbacks [48]; that is, the extended entity or attribute can only be defined by text, which makes data interoperability difficult and cannot use standard tools to perform a consistency check [38]. The ADE mechanism always generates corresponding XML schema files in a separate namespace, which has no impact on data interoperability and consistency checking. Therefore, this method has become the first choice of developers.

These new feature classes are associated with existing CityGML classes through so-called ADE-hooks, which are described in the CityGML specification. An ADE schema may be developed manually, or automatically derived from a UML model based on the transformation rules described in van den Brink et al. [49]. The CTWLADE is documented as a UML diagram.

## 4.2. Overview of the Data Model

According to Section 3.1, the simulation process of waterlogging can be divided into three relatively independent processes: Precipitation, ground surface runoff yield and confluence, and drainage network confluence. Therefore, this paper combines the CityGML ADE mechanism to expand the model in the waterlogging domain. The three modules of Weather, Subcatchment, and PipeNetwork (PipeNetworkSystem and NetworkFeature) modules are designed to correspond to the three simulation processes to save the related entities and properties. Besides, to visualize the flood routing process and support the storage of dynamic information, WaterBody and TimeSeries modules have also been designed. The relationship between these five modules is shown in Figure 2.

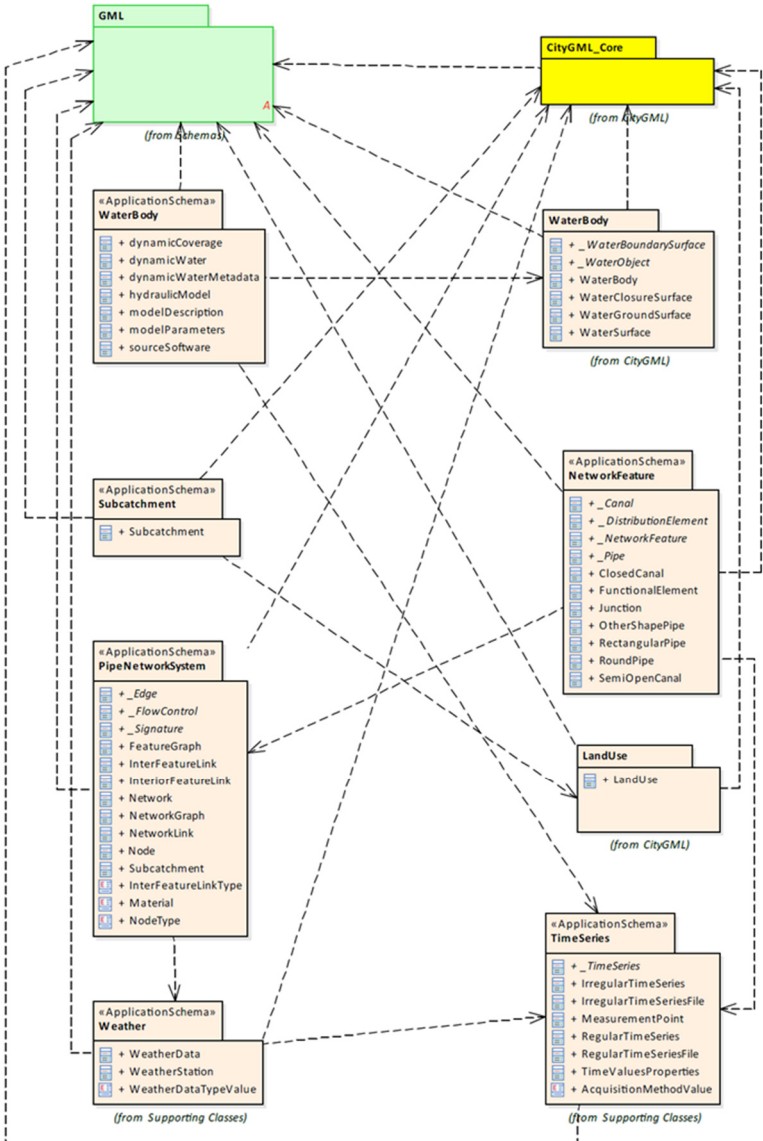

**Figure 2.** UML (Unified Modeling Language) model module structure diagram of the CTWLADE (City Waterlogging Application Domain Extension).

(1) The pipe network module consists of two parts: The PipeNetworkSystem represents the urban drainage network system, and the NetworkFeature represents a series of physical elements in the drainage network;

(2) The Subcatchment module integrates various ground surface data and is a necessary input object for waterlogging simulation;

(3) The Dynamic waterbody module defines several classes related to the position and depth of the water body, which are used to represent the dynamic information related to the accumulation of water on the ground surface;

(4) The TimeSeries module is an important supporting module. All dynamic information in the CTWLADE depends on the module for recording;

(5) The Weather module is mainly used to record meteorological data such as temperature and precipitation and provide support for waterlogging simulation.

### 4.3. Pipe Network Module

Drawing on UtilityNetworkADE [50], the pipe network module is designed to provide a variety of classes and concepts to simulate urban drainage networks, embedding three-dimensional urban drainage network systems into three-dimensional virtual urban environments and correlating them. The module consists of two main parts: PipeNetworkSystem and NetworkFeature.

### 4.3.1. UML Model Design of PipeNetworkSystem

PipeNetworkSystem is the core part of the urban drainage pipe network module, which mainly represents the topological relationship of the drainage pipe network. The UML class diagram is shown in Figure 3. The base class of this part is the abstract class _NetworkFeature, which is a subclass of _CityObject in the core module of CityGML. It is a further conceptual abstraction of semantics and theme classification of entities in the drainage network system. It is used to establish the links between the objects and the aboveground/underground drainage network system. A collection of multiple _NetworkFeature instances can be combined into a drainage network, and the network can be disassembled into multiple sub-networks by self-aggregation, which is convenient for expressing the hierarchical structure of the urban drainage network system. Similarly, each _NetworkFeature may also contain other _NetworkFeatures, which are represented in UML class diagram by a self-aggregation method called consistof. On the one hand, this method improves the flexibility of the model expression, on the other hand, it can also simulate the hierarchy and subordination between the entity elements of the drainage network in detail.

### 4.3.2. UML Model Design of NetworkFeature

PipeNetworkSystem gives the overall structure, logic, and topological relationship of the drainage network, but cannot express the geometric and semantic information of the physical elements in the drainage network. Therefore, this model further refines the NetworkFeature and assigns specific semantic information to integrate the corresponding waterlogging-related multi-source data to meet the requirements of the waterlogging simulation. The UML class diagram is shown in Figure 4.

According to the data related to the drainage network system in the waterlogging domain summarized, the DistributionElement needs to define static properties, such as the name, the upstream and downstream nodes (upStreamNode and downStreamNode), the roughness coefficient (roughnessCoefficient), the length (length), and the maximum depth (maxDepth), etc., as well as dynamic flow (flowSeries) to represent the dynamic drainage state in the drainage network (this type of property is TimeSeries timing type, can record a series of time-related state values, details of TimeSeries in Section 4.6). According to the existing drainage facilities, the DistributionElement can be divided into two categories: _Canal and _Pipe. _Canal can be further refined into closed channels (ClosedCanal) and semi-open channels (SemiOpenCanl). The _Pipe class defines the initial flow (initialFlow), maximum flow (maxFlow), inflow loss (inffluentOffSet), and outflow loss (effluentOffset) required in the waterlogging simulation. Depending on the shape of the drainpipe, _Pipe can be further divided into RoundPipe, RectangularPipe, and OtherShapePipe. Based on these differences,

specific properties are further defined, such as the diameter of the RoundPipe (interiorDiameter) and the height and width of the RectangularPipe (interiorHeight and interiorWidth).

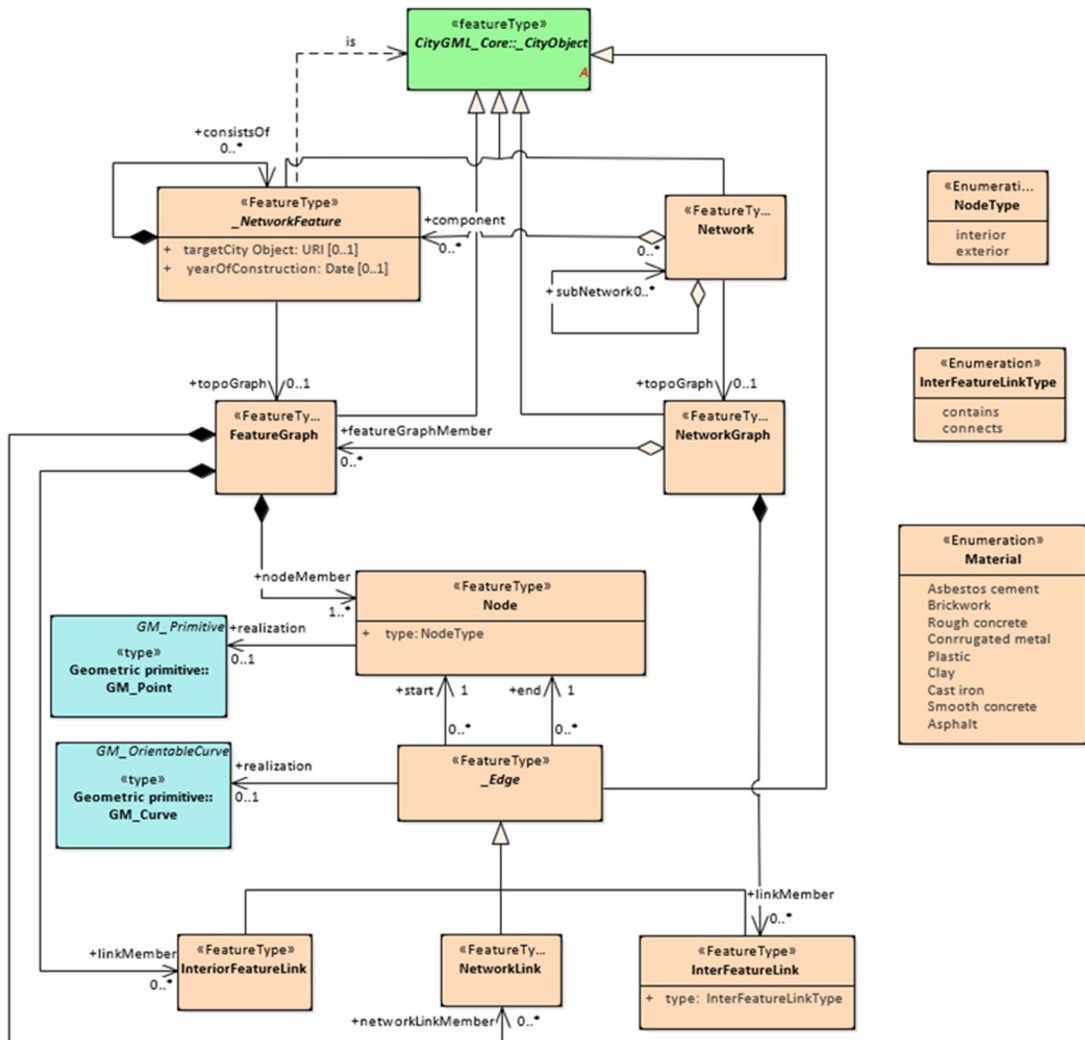

**Figure 3.** UML (Unified Modeling Language) class diagram of PipeNetworkSystem.

FunctionalElement mainly represents the functional entities such as inspection wells, rainwater rafts, and reservoirs in the urban drainage system. Therefore, according to the data related to the drainage system in the waterlogging domain, the FunctionalElement class defines the name, description, position, and invert elevation, and so on. Inspection wells that play an important role in the waterlogging simulation process are individually defined as Junctions and inherit the FunctionalElement class. The Junction class defines the unique property information of the inspection well, such as the first overflow time (firstOverflowTime), the maximum depth (maxDepth), and the dynamic overflow (overflowTraffic). These attributes ensure that the data model can monitor the state of the well in real-time. At the same time, it can also provide real-time dynamic data for subsequent groundwater accumulate simulation calculations.

### 4.4. Subcatchment Module

The subcatchment is an important basis for distributed simulation of waterlogging models and is the smallest unit for simulation of ground production and confluence. The subcatchment is a spatial area defined by the terrain and drainage pipeline nodes. It contains all the ground surface data of the

area and is a natural data integration object. Therefore, the Subcatchment module is designed in this model, and its UML class diagram is shown in Figure 5.

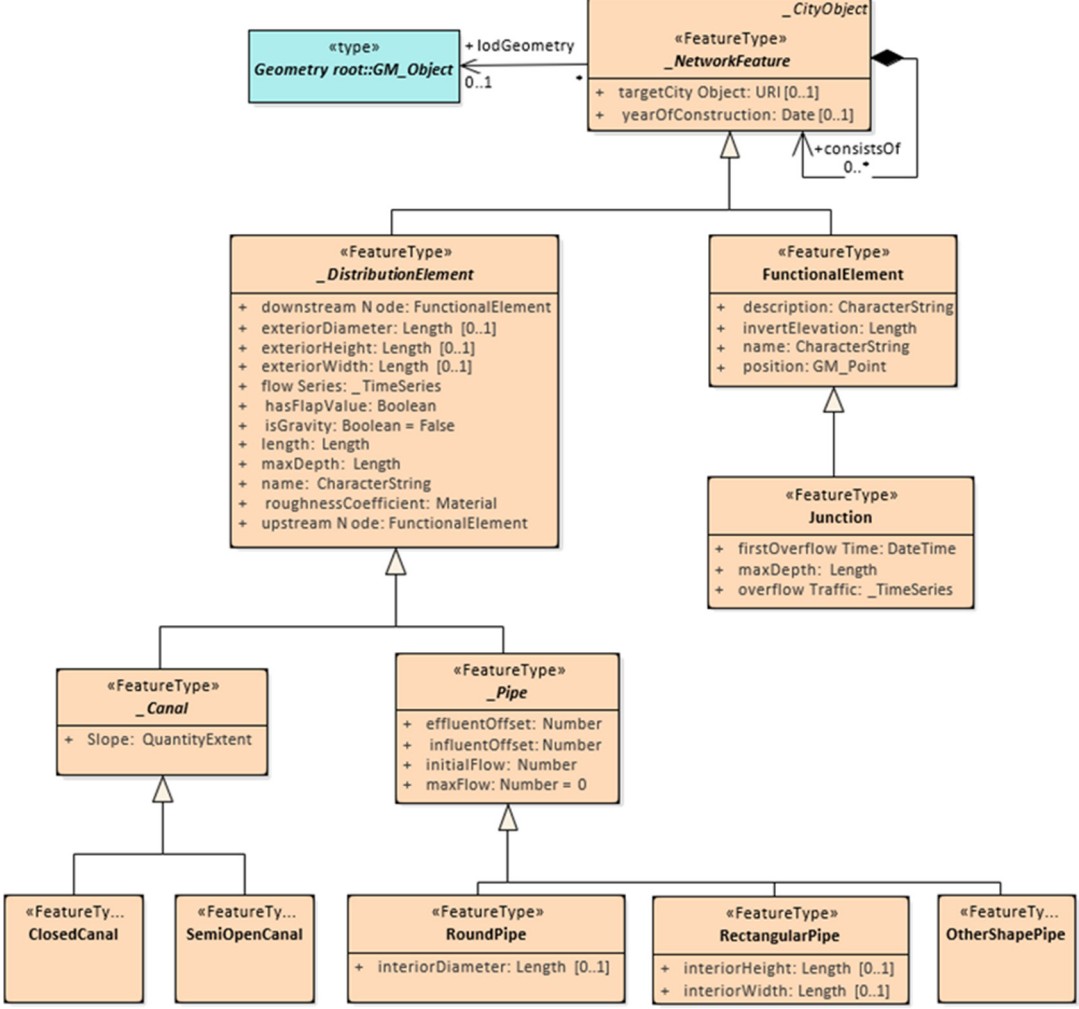

**Figure 4.** UML class diagram of NetworkFeature.

The Subcatchment class inherits the _CityObject class from the CityGML core module and is used to record the ground surface data required in the waterlogging simulation, including name, area, imperv, and roughnessOfImperv, slope, width (feature width), etc. The area and roughness of the permeable area/impermeable area are determined by the land use type of the subcatchment. Therefore, the LandUse class of the land use theme module in CityGML has a relationship with the Subcatchment class. Furthermore, each subcatchment corresponds to a node of the drainage network as the outlet of the confluence, so the Junction class is connected to the Subcatchment class through a one-to-many association called support. For the large-scale waterlogging simulation, the impact of the rainfall in each subcatchment is not the same. Therefore, it is necessary to match a set of meteorological data for each subcatchment. This module uses the association to express the relationship between the WeatherStation class and the Subcatchment class. Finally, to express the geometric information, the sub-catchment is defined as the GML MultiSurface type in the LoD0-4 level, which has three-dimensional coordinate information and can also be assigned different colors, which is the basis for visualization of multi-state subcatchment.

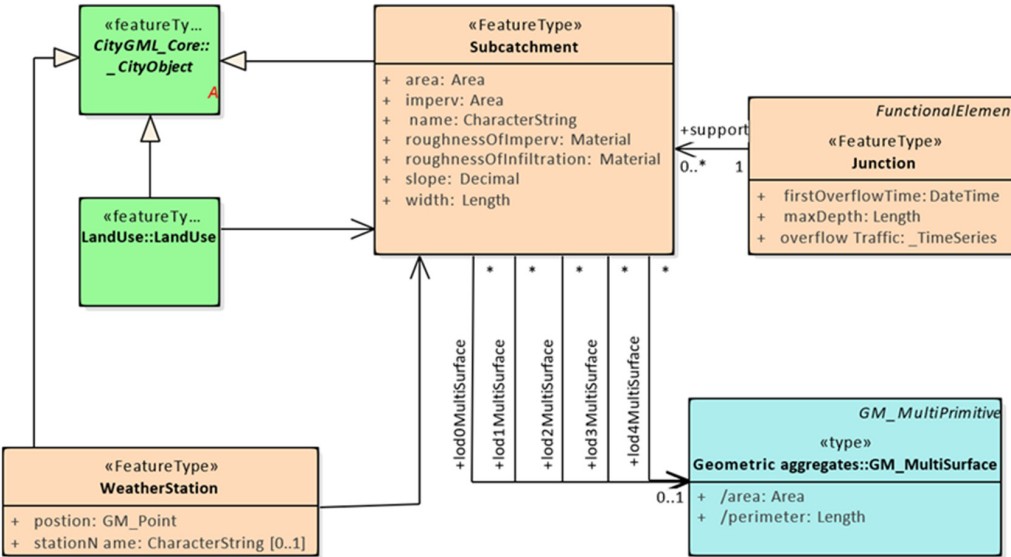

**Figure 5.** UML class diagram of Subcatchment.

### 4.5. Dynamic waterbody Module

Dynamic water accumulation information is one of the important information in the simulation results of waterlogging models, and it is also an important basis for urban planning and disaster relief decision-making. To be able to integrate dynamic water accumulation information and realize the 3D dynamic of this information, the dynamic waterbody module, WaterBody, is designed in this model. The UML class diagram is shown in Figure 6.

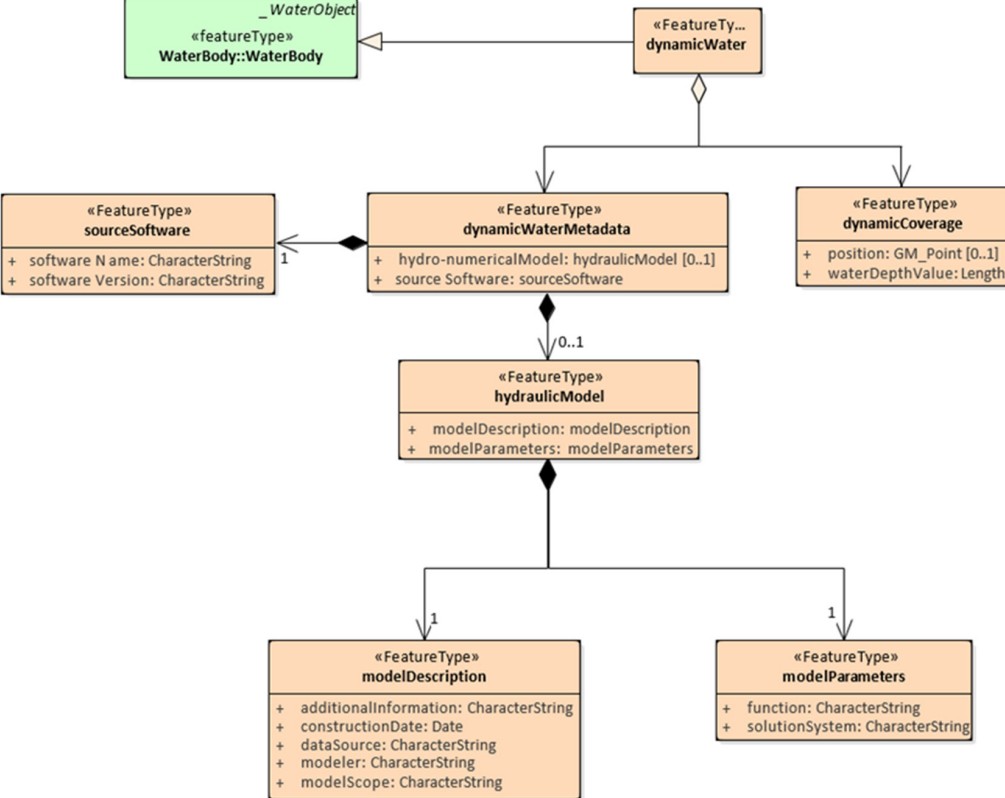

**Figure 6.** UML class diagram of WaterBody.

The base class of the WaterBody module is the dynamicWater class, which inherits the WaterBody class from the CityGML water theme module to record and express all water-related information in the waterlogging simulation results. According to the type of information stored, dynamicWater is divided into two categories: Metadata (dynamicWaterMetadata class) and dynamic water body (dynamicCoverage class).

Metadata is an important piece of information for backtracking and evaluating datasets from different waterlogging models. In this model, the metadata is divided into two parts: Source software information (sourceSoftware class) and waterlogging model information (hydraulicModel class). The sourceSoftware class is a necessary attribute of the dynamicWaterMetadata class and records the software name (softwareName) and software version (softwareVersion). This information can help users understand the model type and data format. The hydraulicModel class is an optional property of the dynamicWaterMetadata class, including the model description (modelDescription class) and model parameters (modelParameters class). The modelDescription class provides information such as the model provider, the model creation date (constructionDate), and the model application domain (modelScope). The modelParameters class provides information about the mathematical basis of the waterlogging model.

The dynamicCoverage class is the core class of the dynamic water module. It records the dynamic range and depth information of the water accumulation by two properties: Position and water depth value (waterDepthValue). Both position and waterDepthValue are _TimeSeries time series type data. Position records the dynamic range of the groundwater accumulation and waterDepthValue records the dynamic water depth information of the water accumulation. These two attributes are related to each other at the same time point to form a three-dimensional dynamic water accumulation during visualization.

## 4.6. TimeSeries Module

To make the data model proposed in this paper meet the dynamic expression requirements of relevant information in waterlogging simulation, such as dynamic flow in drainage pipeline, dynamic overflow information of inspection well, dynamic diffusion information of ground flooding, etc., in this model, the time series module, TimeSeries, is designed to record a series of state values related to time. The UML class diagram is shown in Figure 7.

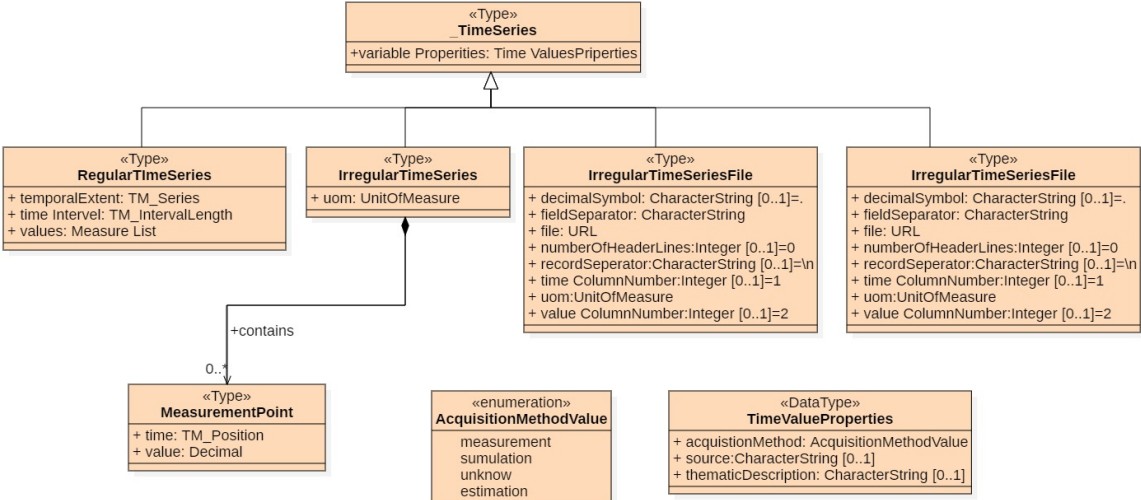

**Figure 7.** UML class diagram of TimeSeries.

To improve the compatibility of data models, time-series data are divided into two categories: Regular time series data and irregular time series data. In the rule time data, two attributes including a period (temporalExtent) and a constant time interval (timeInterval) are defined, and the regular time

series data values are saved as a MeasureList list, which corresponds to the regular time nodes one by one. The regular time series data itself can be stored directly in an XML document (RegularTimeSeries class), or an external file (RegularTimeSeriesFile class) with a table structure. The RegularTimeSeriesFile class defines easy to read the time-series data, such as a field separator (filedSeparator) and a file address (file), the offset of first line (numberOfHeaderLines), record separator (recordSeparator) and other attributes. In irregular time-series data, each value has a corresponding timestamp. Similarly, instances of irregular time series data can be stored in an XML document (the IrregularTimeSeries class that contains the MeasurementPoint object) or in a table-based external file (IrregularTimeSeriesFile class). All specific time-series data types have a common set of metadata (variableProperties) whose data type is TimeValuesProperties, which is used to specify related description information of time series data, such as data source, theme description (thematicDescription), or data acquisition method (acquisitionMethod). The enumeration type AcquisitionMethodValue gives several common data acquisition methods, such as measurement, simulation, estimation, and so on.

### 4.7. Weather Module

Time-related weather or meteorological data is essential for waterlogging simulation. To integrate the corresponding meteorological data in this model, the WeatherStation and WeatherData are designed and used, and the UML class diagram of the Weather module is shown in Figure 8.

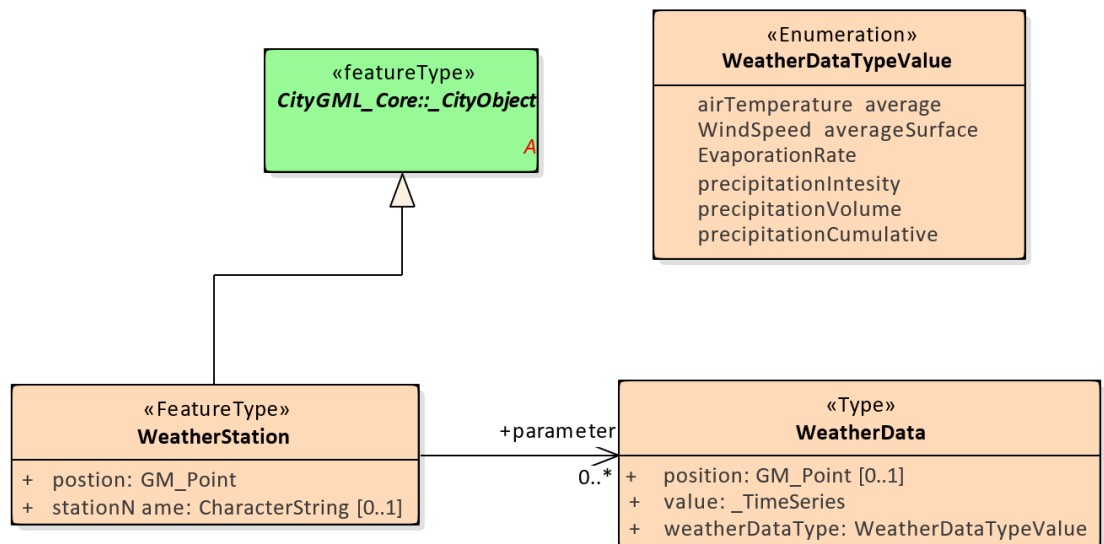

**Figure 8.** UML class diagram of Weather.

The WeatherDate class records a series of time-correlated weather data via the _TimeSeries type, and the type of weather data (weatherDataType) is given by the enumeration type WeatherDataTypeValue. The WeatherDataTypeValue defines the average ground surface evaporation rate (averageSurfaceEvaporationRate) and the rain gauge required in the waterlogging model. Generally, the rain gauge can be expressed in three ways, so the precipitation intensity, the precipitationVlolume, and the precipitationCumulative are defined here. It is worth noting that the averageSurfaceEvaporationRate is sometimes not directly given in the data source and needs to be calculated by the empirical formula. The calculation of the averageSurfaceEvaporationRate is related to the air temperature (airTemperature) and the average wind speed (averageWindSpeed), so they are also defined in the WeatherDataTypeValue. Each city object can be associated with any number of WeatherData objects, or it can be directly associated with WeatherStation that aggregates different WeatherData objects to provide data support for waterlogging simulations. At the same time,

the name and location of the weather station are defined in the WeatherStation class, which facilitates the retrieval and location of information and spatial analysis.

## 5. Method of Data Integration, Sharing, and 3D Visualization

### 5.1. Method of Data Integration

#### 5.1.1. Data Integration Method Based on FME Object

For buildings, the bottom profile data of two-dimensional buildings may have elevation-related attributes such as floor height or floor number, which can be used to directly extend the three-dimensional solid model in FME by extrude. However, these two-dimensional building surface data do not have the bottom elevation information, which cannot directly form the building distribution scene in line with the terrain fluctuation. Therefore, when building the solid model of this kind of object, it is also necessary to assign the corresponding building bottom elevation according to the 3D terrain formed by DEM, that is, the surface Draper in FME, so that the 3D model fits the 3D terrain distribution.

The urban drainage network data are different from the basic geographic data. Although it does not have the precise and detailed elevation information like the laser point cloud, it also has certain elevation attributes, which can be directly used for 3D modeling. The data of urban drainage network can be divided into two categories: Pipeline data and network node data. For pipelines, they have the attributes of pipe diameter (or width and height) and elevation at upstream and downstream nodes. Therefore, according to these attribute data, 3D buffer and 3D force can be used to process 2D linear pipeline data to form 3D tubular drainage pipeline model. For the pipe network node, it has two attributes of bottom elevation and well depth, so it can directly use extrude for 3D stretching to form a 3D pipe network node model. The processing flow is shown in Figure 9.

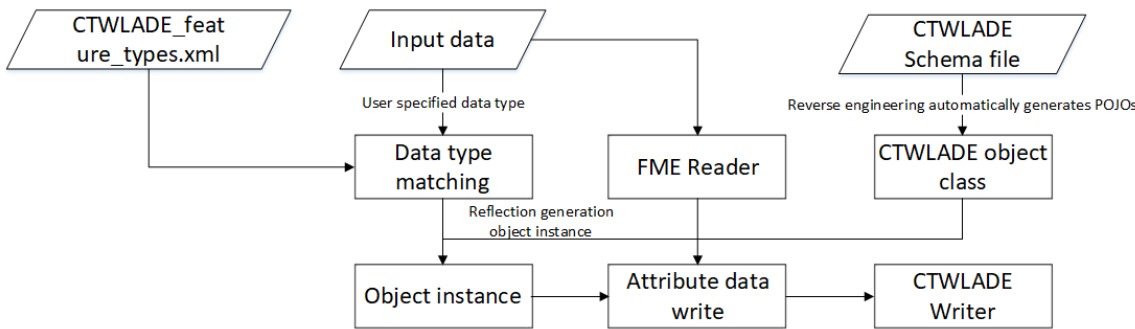

**Figure 9.** Attribute data integration process.

#### 5.1.2. Data Integration Method Based on Citygml4j

The dynamic time series data in the CTWLADE model mainly include the dynamic meteorological data, the dynamic flow data of drainage pipe network, and the dynamic ponding data on the ground. The semantic entities of these data have been integrated into the CTWLADE model in advance. For example, the semantic entities of the dynamic meteorological data meteorological stations will be integrated into the CTWLADE model in FME as part of the input data of the urban waterlogging model. Therefore, when integrating these dynamic data, we need to determine their respective semantic subjects first. When citygml4j is used to operate on CityGML data, all objects in CityGML data will be deserialized into a CityGML data tree according to the semantic level_ Model is the object tree of the root node, and its operation flow is shown in Figure 10.

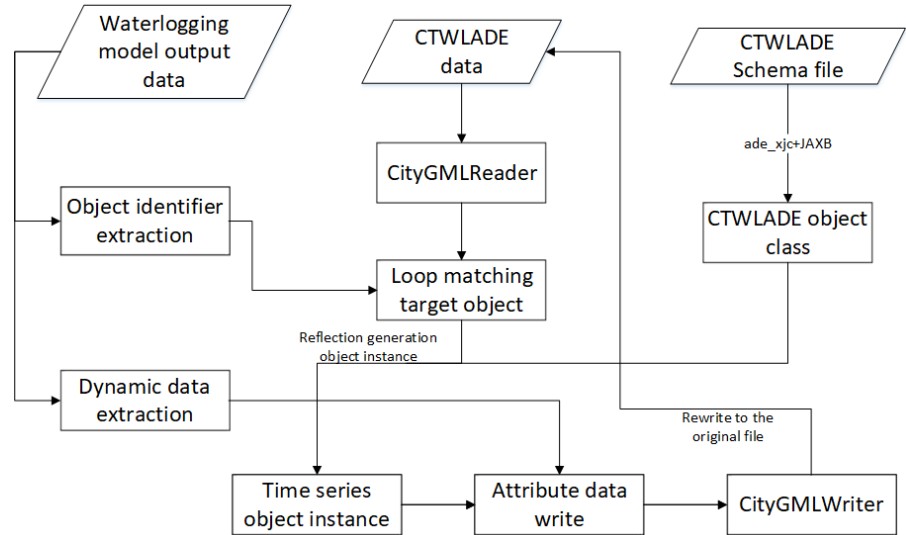

**Figure 10.** Integration process of dynamic time series data.

### 5.2. Method of Data Sharing

One of the characteristics of CTWLADE model is that it defines a large number of objects and attributes related to pipe network state and ground water accumulation, which provides the possibility for 3D dynamic visualization of urban waterlogging scene. Based on the characteristics of all kinds of 3D model formats, a method of visual data organization and sharing of KML/glTF dataset is proposed.

KML has the advantages of simple structure and easy analysis, and it can also save two attribute information through < placemark > and < name > tags; however, the texture information of three-dimensional model cannot be saved in the process of transforming from CWTLADE model to KML, so KML itself is not suitable for this article's choice in three-dimensional visualization on the web. The latest three-dimensional graphics format glTF can perfectly maintain all texture details in the CWTLADE model, and support model monomer and tile storage, but glTF format cannot contain any attribute information. The organization of KML/glTF dataset is shown in Figure 11.

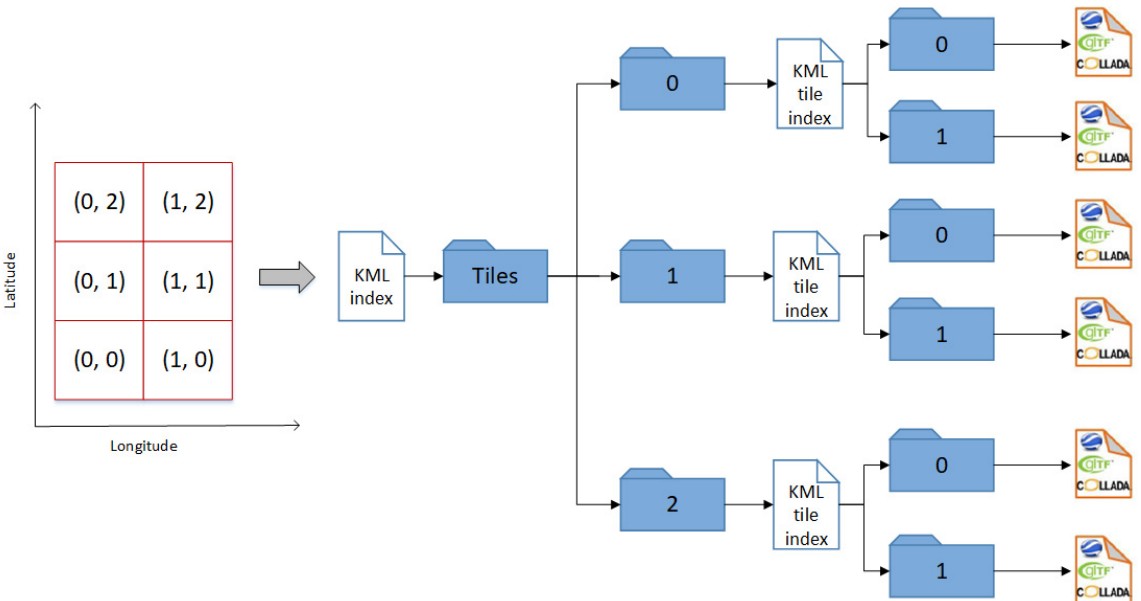

**Figure 11.** Organization form of KML/glTF dataset, taking 2 ∗ 3 tiles as an example.

While providing geometric information of 3D objects, according to the unique identifier GML_ ID retrieves the CTWLADE model data and organizes the corresponding attribute information in the form of JSON. Gltf files are also described in JSON format, so they can be published and shared through geoserve. The data sharing process for 3D dynamic visualization is shown in Figure 12.

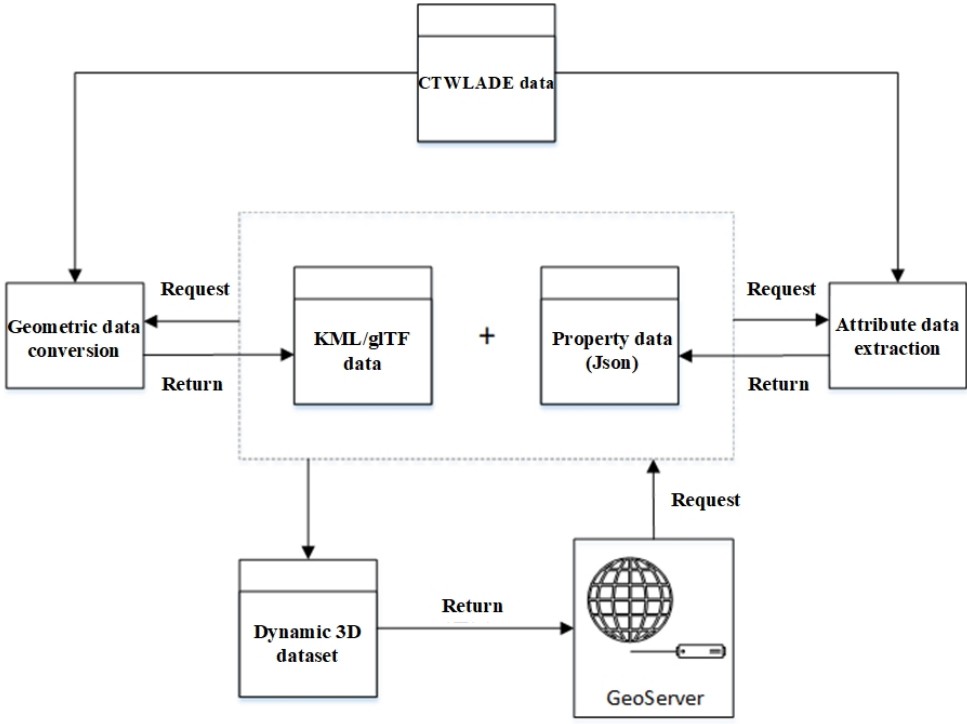

**Figure 12.** Data sharing process for 3D dynamic visualization.

### 5.3. Method of 3D Visualization

(1) Thre-dimensional visualization method of basic information.

In this paper, an explicit link between 3D visualization model and attribute data is proposed, in which geometric model and attribute information are associated in 3D network client. The JSON file of attribute data are uploaded to the online JSON parser for storage, and then published through Google cloud. The corresponding procedure flow is shown in Figure 13.

The online JSON parser can abstract JSON data into a database table-like structure, with the first row defining the attribute name and the other rows storing the corresponding attribute values of each 3D object. The logical link between the 3D model and the corresponding row is established by the gml_id column in the table structure, which contains the unique identifier of the 3D object. Each of the remaining columns represents an attribute of a 3D object. Google fusion provides an API that can execute SQL statements, so GML saved in KML can be saved in 3D network client_ ID, as the key value of query, gets all the corresponding attribute information from the online JSON parser. By using the free Google cloud hard disk application, all users who have access to the online JSON parser can edit it to keep the content up to date without affecting the original geometry state. In addition, separating attribute data from 3D visualization model has the following advantages: Any update of attribute content can only occur in JSON file, so there is no need to export and deploy 3D visualization model again.

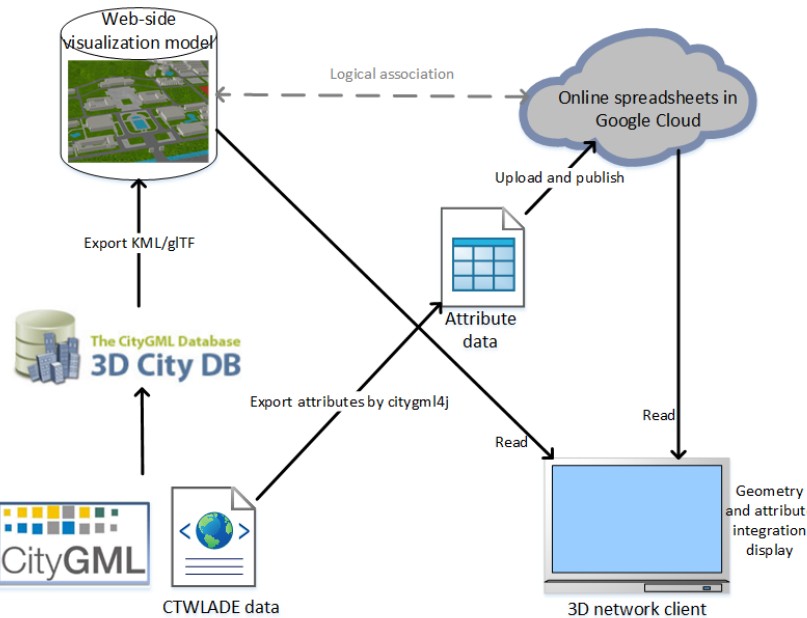

**Figure 13.** 3D visualization process of integrating attribute information.

(2) Attribute-driven 3D dynamic visualization method.

There are two kinds of dynamic data that need to be visualized in CTWLADE model, one is the dynamic displacement data of each drainage pipeline after waterlogging simulation, and the other is the dynamic ponding data on the ground. The first mock exam is the visualization of dynamic attributes for the same model, while the latter is loading different independent objects by time. Both of them depend on their own attribute information.

For the pipeline model, the corresponding online JSON parser records the drainage time series data of each pipeline model, which can be used as key value pair, key as time and value as drainage. Therefore, the displacement can be mapped to different colors to achieve the effect of superposition display of drainage information and 3D spatial model. For example, for a polygon entity, you can set its material attribute to achieve a variety of rendering effects such as color, picture, and grid filling, as shown in Figure 14. Then, the time listener is set in cesium. When the time reaches the key, the pipeline model is rendered by setting the color corresponding to value, so as to realize the dynamic visualization of the drainage of the pipeline model.

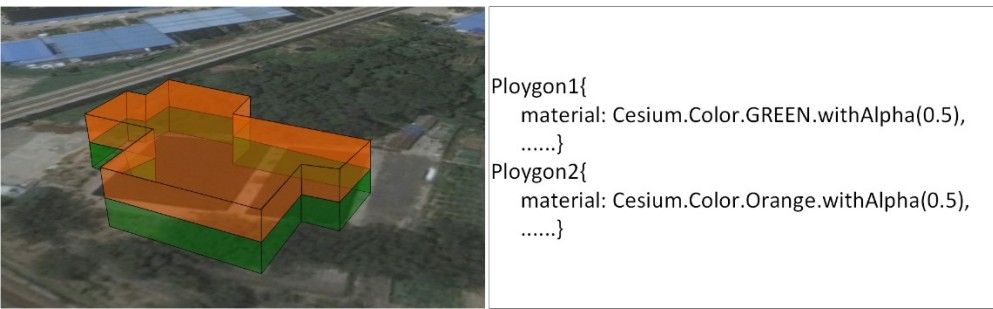

**Figure 14.** 3D visual color fill diagram and code.

For dynamic water logging data, different 3D water logging models need to be loaded at different time points, so in this paper, the idea of tile map is used to slice the 3D water logging model on time scale to form time-series tiles. Time-series tiles refer to the existence of multiple sets of tile data; each set of tiles is not independent but rather related to time. In cesium, you can load the model at the URL (uniform resource). In the case of locator, the specific time folder name is replaced by {times},

and then the models under different time folders are loaded according to the time axis by using the timeintervalcollectiong class and datacallback function, so that the dynamic visualization of 3D surface water can be realized with the change of time axis.

## 6. Experiment Verification

To verify the validity of the CTWLADE, the prototype system uses CTWLADE as the data medium to integrate and share data for the SWMM (storm water management model, one of the most widely used waterlogging models). The functional modules of the system are mainly divided into three parts: Multi-source data processing and integration, data sharing, and three-dimensional interactive dynamic visualization. The processing flow of the prototype system is shown in Figure 15.

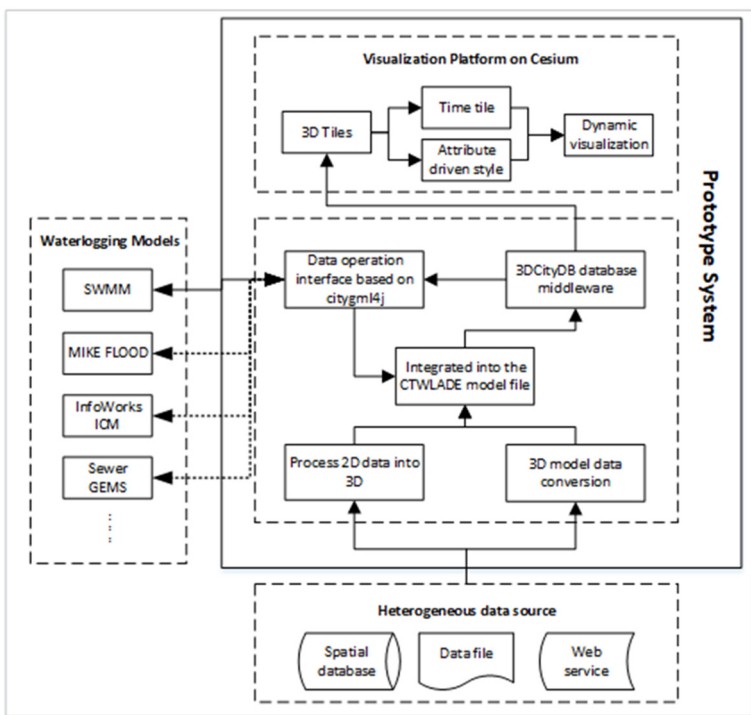

**Figure 15.** The processing flow of the prototype system.

### 6.1. Software Development Environment

The system is built on the laboratory desktop computer, and the specific hardware environment is as shown in Table 2.

**Table 2.** Hardware environment configuration of experimental platform.

| Computer Model | HP Compaq 8380 Elite MT |
|---|---|
| CPU | Intel i7-3770 3.4 GHz |
| RAM | 16 G |
| Hard disk | 1 T |
| Operating system | Windows 10 64 bit |

The related software involved in the research process of this paper includes JDK, PostSQL, Tomcat, etc., as follows:

Integrated development environment (IDE): IntelliJ idea;

Development languages: Java, JavaScript;

Java Software Development Kit: JDK 1.8;

Spatial database: PostSQL 9.6;
Development of operating system: Windows 10;
Web server: Tomcat 8.0.41.

*6.2. Study Area and Data*

Based on the prototype system, this paper selected the Xianlin Campus of Nanjing Normal University in China as the experimental area and collected the multi-source data required for the waterlogging simulation, as shown in Table 3. All spatial geographic data are unified to the WGS 1984 UTM Zone 50N (EPSG: 32650) coordinate system.

**Table 3.** Experimental area basic data summary table.

| Data Type | Data Format | Data Accuracy | Included Attributes |
|---|---|---|---|
| Precipitation data | Excel | 10 min | Time, Precipitation |
| DEM | GeoTiff | Resolution 10 m | Elevation |
| Main building data | Shapefile | Scale 1:1000 | Name, Floor |
| Drainage pipe | Shapefile | Scale 1:1000 | Pipe number, upstream and downstream node number, pipe start height, length, pipe diameter, material |
| Node | Shapefile | Scale 1:1000 | Node number, bottom elevation, depth |
| Land use | Shapefile | Scale 1:1000 | Land use type, area |

(1) Precipitation data.

The precipitation data come from the meteorological service center of Jiangsu Province, which is measured by the Xianlin meteorological station of Nanjing Normal University in Xianlin Campus. The table structure is shown in Table 4. The data are stored as an excel file, which records the precipitation data between 00:00 and 06:10 on August 17, 2018. The data sampling interval is 10 min. Each line of the document includes station code, precipitation collection time, and precipitation data. The precipitation data is the accumulated precipitation (unit: mm) from the last non-zero numerical sequence.

**Table 4.** Rainfall data format description and examples.

| Station Code | Acquisition Time | Accumulated Precipitation |
|---|---|---|
| M3554 | 201808170500 | 2.4 |
| M3554 | 201808170510 | 0.7 |
| M3554 | 201808170520 | 1.2 |
| M3554 | 201808170530 | 3.4 |
| M3554 | 201808170540 | 2.0 |
| M3554 | 201808170550 | 0.4 |
| M3554 | 201808170600 | 0.1 |

(2) DEM data.

The DEM data storage format used in this paper is GeoTIFF, and the grid size is 10 m × 10 m, as shown in Figure 16a.

(3) Main building data.

The main building data are the footprint data of the main buildings in Xianlin Campus of Nanjing Normal University, which is digitized from the 1:1000 topographic map of Xianlin Campus of Nanjing Normal University, and the storage format is Shapefile. This data is mainly used for 3D modeling, so each building has corresponding floor number attribute, as shown in Figure 16a.

(4) Drainage pipeline data.

The data of drainage pipeline are from the field measurement of professional engineering surveying and mapping company, mainly including the Shapefile line file of pipe section and Shapefile point file

of node. The pipe section data include the number, pipe diameter, length, material, elevation of the beginning and end of each pipe section, and the node data include the depth of the rainwater well, inspection well, water outlet, etc., and the bottom elevation. These data will be integrated into the CTWLADE model, and directly face the urban waterlogging model. According to the requirements of the waterlogging model, some generalizations are carried out for them, and the specific results are shown in Figure 16a.

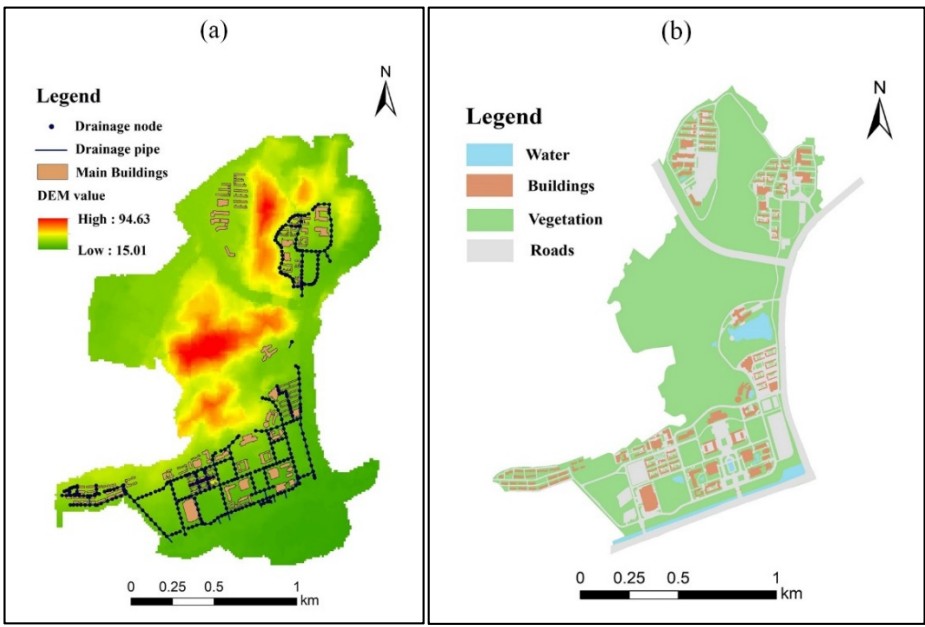

**Figure 16.** Visualization of raw data in experimental area. DEM data (**a**) land use data (**b**).

(5) Land use data.

The land use data are digitized from the 1:1000 topographic map of Xianlin Campus of Nanjing Normal University, and the storage format is Shapefile file. Considering that the land use data are mainly used to calculate the area of permeable area and impervious area in waterlogging simulation, the land use is not divided into building land, road, green land, and water body, as shown in Figure 16b.

*6.3. Results and Analysis*

The data integration processing module that imports the experimental data in the background will generate the CTWLADE data at LOD1 of the research area. For further visualization, the corresponding KML/glTF dataset (which can be sported by 3DCityDB-Web-Map-Client, see https://github.com/3dcitydb/3dcitydb-web-map for details) is also processed. The visualization results of loading model data into the prototype system is shown in Figures 17 and 18. The 3D virtual environment of the study area includes basic data on the ground, such as 3D buildings, terrain, land use, sub catchments, etc., and drainage pipeline systems, such as drainage pipes and nodes.

The prototype system can export the data required by the SWMM based on the CTWLADE data, as shown in Figure 19. The output data are a text file in *.inp* format, which contains the research area range, the simulation start and stop time, the simulated time step, and information of the drainage pipe, the node, the Subcatchment, etc. If some of the waterlogging simulation parameters are not recorded in the integrated data, they will be written into the exported data according to the experience value recommended in the SWMM operation manual. The data exported by the prototype system can be directly imported into the SWMM for waterlogging simulation.

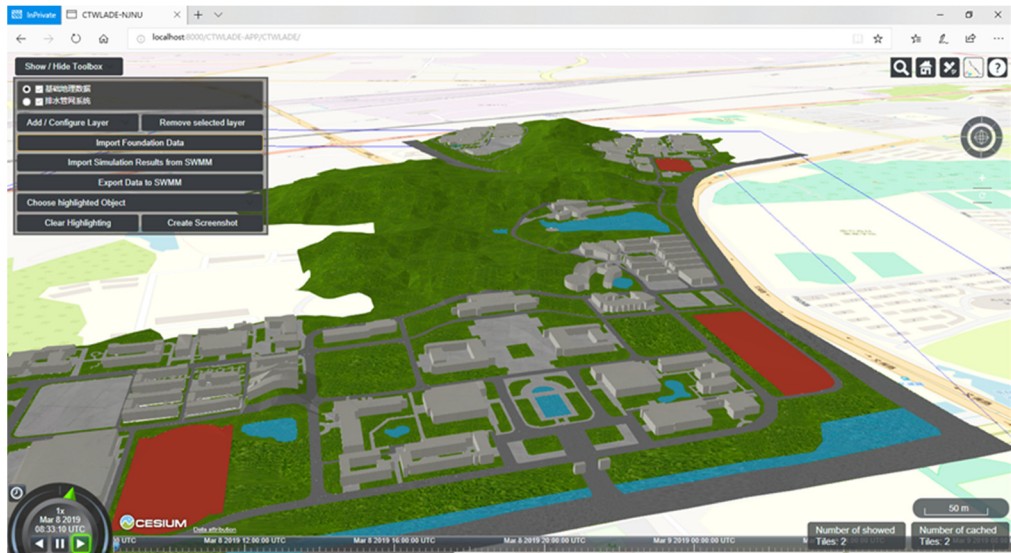

**Figure 17.** 3D virtual environment of research area generated based on city waterlogging application domain extension (CTWLADE) data.

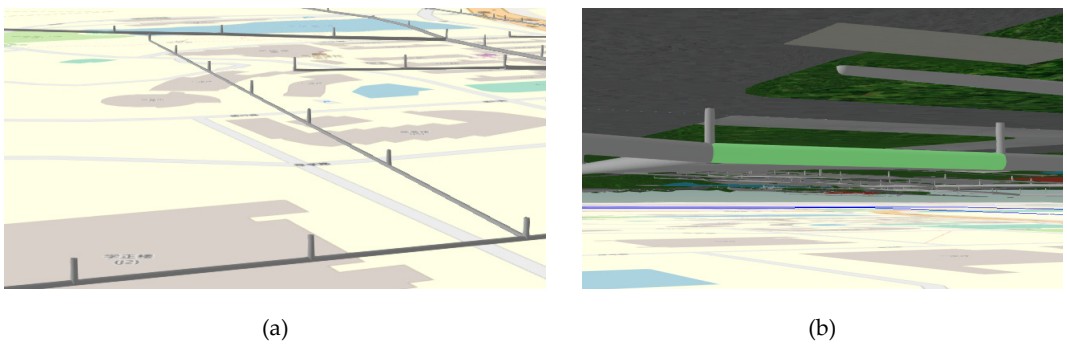

(a)                                                                                  (b)

**Figure 18.** 3D drainage pipeline system in the study area: (**a**) Visualization of 3D drainage pipeline system alone; (**b**) 3D virtual environment of the study area from the underground perspective.

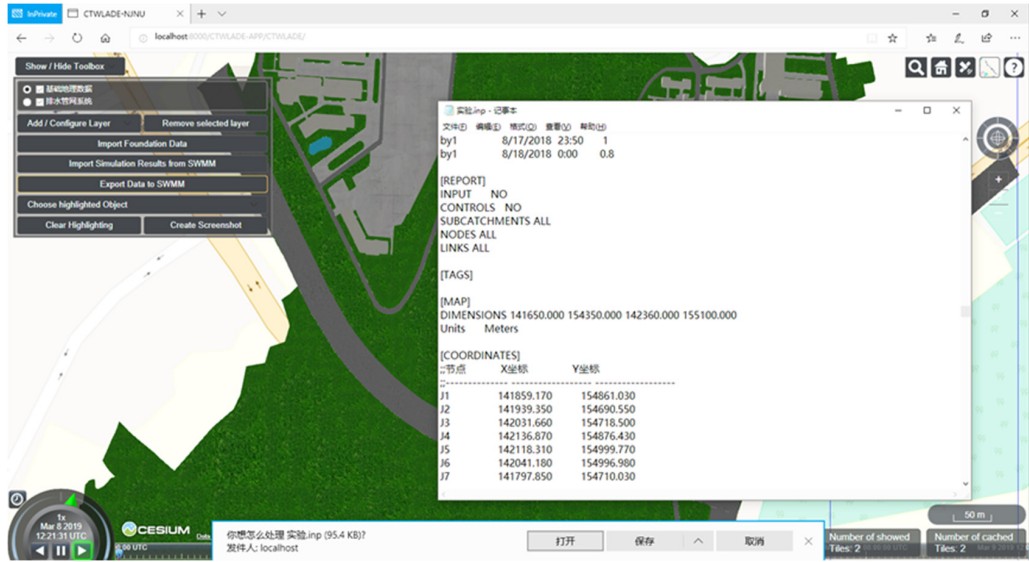

**Figure 19.** Storm water management model (SWMM) input file derived from CTWLADE of the study area in the prototype system.

SWWM simulation results can give information about overflow volumes, so the flood coverage is calculated based on the node overflow in the SWMM simulation results. From the perspective of active diffusion, an active water accumulation diffusion algorithm for dynamic distribution of water quantity is proposed [51]. All potential water accumulation points in the study area are set as diffusion sources. During the simulation of waterlogging, the amount of water accumulation in each diffusion source is calculated first, with the grid of diffusion sources as the center. Through the trial algorithm, the water quantity diffuses outwards in circles, and the elevation of water accumulation is carried out in the diffusion process (elevation of ponding grid value plus waterlogging depth) compared with the elevation value of the adjacent outer circle grid until the elevation value of the outer circle is greater than the waterlogging elevation. Finally, the shapefile data of ponding range are extracted, and the attributes include water level.

Firstly, the shapefile data of the ponding range are integrated into the CWTLADE model of which the method is provided in Section 5.1. Then, the dynamic visualization method of water accumulation is mentioned in Section 5.3. An important module is time series, which loads different 3D water accumulation models at different time points. Figure 20 shows the waterlogging process after the overflow of the 1Y8 node on the Nanjing North Road in the North District of NNU of four time steps (10 min) of ground water accumulation between 05:40 and 06:10 on August 17, 2018. The range and depth of water accumulation are mainly determined by the depression and height difference of underlying surface. We can see that the range of water accumulation is increasing every 10 min. Because the elevation fluctuation of 1Y8 node is not very large, the water level shown in the results do not indicate obvious change. This node is located in the road area between the dormitory building and the playground, which is an important way for students to walk, so waterlogging has a considerable impact on pedestrians.

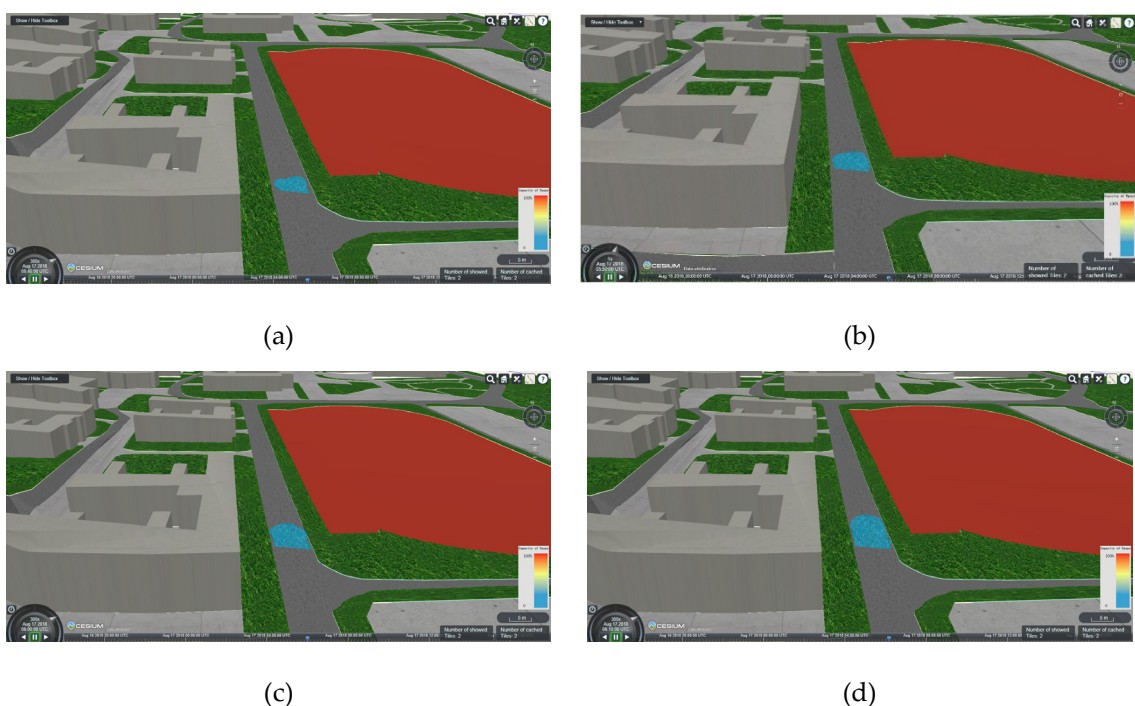

(a)　　　　　　　　　　　　　　　　　　　　(b)

(c)　　　　　　　　　　　　　　　　　　　　(d)

**Figure 20.** The 3D dynamic flooding in steps of ten minutes. (**a**–**d**) are the flooded areas of the road every 10 min between 05:40 and 06:10 on 17 August 2018.

The color of the pipeline shown in Figure 21 ranges from blue to red, indicating that the flow load of pipeline drainage (the ratio of the current flow to the maximum preset flow) is from 0% to 100%. It can be seen that the flow load of the drainage pipelines, which are in the south of the North District of NNU and near the stadium, is very large. As the rainfall continues, the flow load has

successively tended to be 100%. The distribution characteristics of the pipeline drainage state are also consistent with the terrain of the North District of NNU, which is northwest high and southeast low. If we can get the real-time rainfall data, we can provide the relevant departments of the school with the visualization of the drainage load simulation of the pipe network, and propose the possible overload nodes of the pipe network, accurately find the potential waterlogging area, and improve the efficiency of maintenance.

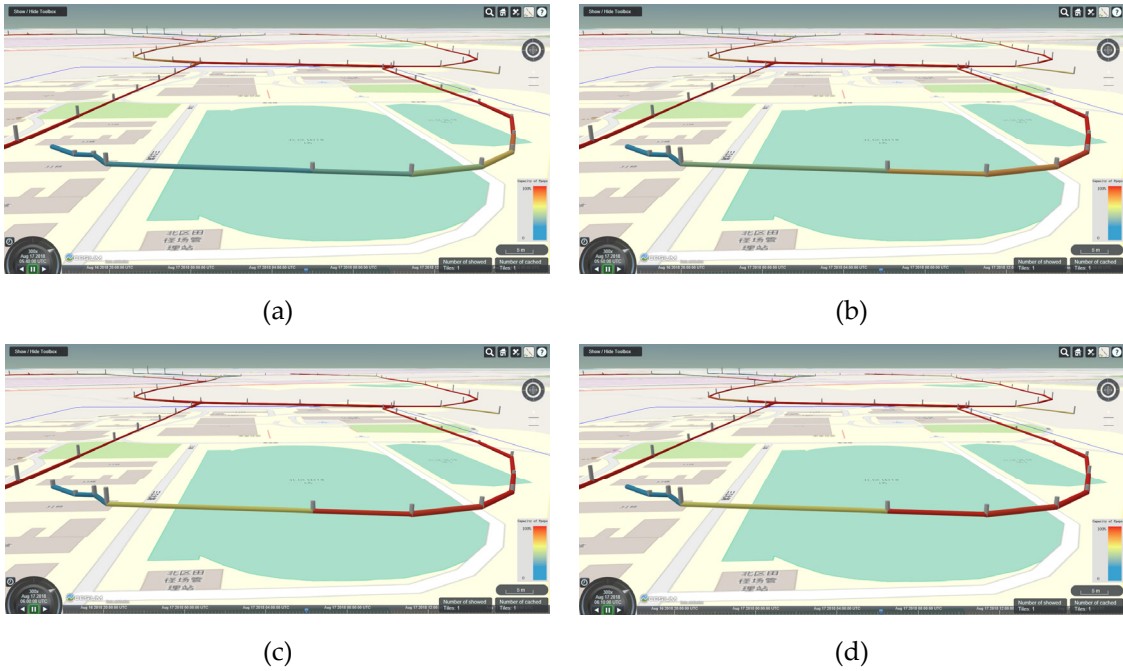

**Figure 21.** The 3D dynamic process of flow of drainage pipelines in steps of 10 min. (**a**–**d**) are the flow load of drainage pipelines for every 10 min between 05:40 and 06:10 on 17 August 2018.

It can be seen from the above process that the data integration and share of waterlogging data based on the CTWLADE model can be used as a bridge to better communicate the data source and the waterlogging model, so that these two aspects can be developed independently without worrying about whether the data source format needs to be compatible with the data requirements of the waterlogging simulation or whether the scholars who study the waterlogging model need to spend a lot of effort on how to integrate and organize the multi-source data. On the other hand, it can provide a data-sharing service that is easy to render and contains attribute information for dynamic 3D visualization, which greatly expands the readability and practicability of the simulation results of waterlogging. It can provide very intuitive and clear data support for disaster management, and it has great help to disaster warning and decision-making.

## 7. Conclusions

With the urbanization process, the urban flooding disasters caused by various reasons have become more serious, and it has become a major "urban disease", causing many inconveniences for urban life and even threatening people's lives and property. To simulate and predict waterlogging, to estimate the risk of waterlogging disasters for old cities to reduce disaster losses, and to provide planning advice for new cities to prevent the occurrence of waterlogging, many scholars have long studied the mechanism of waterlogging stagnation and development and given various waterlogging models, which can perfectly meet the above requirements. However, most of these models require a large amount of data to support, which come from hydrology, meteorology, planning, mapping,

and other departments. The incoordination of space–time scale and format standards has brought great obstacles to the study of urban waterlogging.

To integrate a large amount of geometric and semantic information of waterlogging, and to display the waterlogging simulation results clearly and intuitively, this paper designs and expands the related entities and attributes in the waterlogging domain based on CityGML and its application domain expansion mechanism to generate the waterlogging-related data integration model CTWLADE. On this basis, data integration and visualization implementation methods are proposed. In the experimental stage, the key parts of the multi-source waterlogging-related data integration prototype system based on the CTWLADE model are designed and implemented, including functions like multi-source data integration, exporting data to specific waterlogging models, integrating waterlogging simulation results, and three-dimensional dynamic visualization. The typical test area data are used to test the relevant functions of the prototype system, and the validity and practicability of the CTWLADE proposed in this paper are verified.

However, the CTWLADE is still limited. In the research of this paper, the mechanism and process of waterlogging are properly simplified, and the integrated data can only meet the most basic waterlogging simulation. However, waterlogging is a very complicated process, including not only ordinary drainage pipeline systems, meteorological conditions, and ground surface information, but also more complex pressure valves, pressure pipes, groundwater, etc., and there is even the need to consider the problem of pollution spread caused by waterlogging. Therefore, the data integration model CTWLADE proposed in this paper is not perfect and needs further study. For example, the scope of water accumulation and the simulation process of water level need to be further discussed with experts in the field of waterlogging. Based on the current model, a waterlogging-related multi-information integration model will be gradually established that can describe underground soil, water layer, surface hydrological information, meteorology, complete drainage network system, and groundwater accumulation information.

In future research, it will also be necessary to focus on the evaluation of the presented 3D virtual environment. In addition to the usability of this interactive 3D visualization itself, the UI (user interface) should be tested, too [52,53]. There is lack of defined principles and clear recommendations for the usability of 3D visualizations. For example, some authors, e.g., [54], argue that 3D visualization is able to present geospatial data to wider audiences, including those with little or no GIS experience. On the other hand, results of other studies [55] suggest that 3D visualization, especially the interactive ones, will be more useful for users with previous experience with 3D visualization and for complex tasks in particular. When testing usability related to the scope of the paper, it is possible to use both expert evaluation methods (e.g., heuristic evaluation [56]) and user testing. In the case of user testing, it is especially appropriate to test the solutions of geospatial tasks, measure, and then analyze the accuracy, speed, and used strategy [57]. These characteristics when working with interactive 3D visualizations can be examined mainly using the eye-tracking methods (described for example by [58,59]) or user logging (e.g., [60]). The abovementioned suggestions will even increase impact and applicability of the developments of the CTWLADE achieved so far.

**Author Contributions:** All authors contributed to the idea and concept of a literature review. Jie Shen conducted a literature review. Jie Shen and Jiemin Zhou defined waterlogging related entities and attributes and extended CityGML in waterlogging domain. Jie Shen and Jingyi Zhou developed a use case study, drafted the manuscript and revised the manuscript. Lukas Herman and Tomas Reznik contributed to the analysis and the compilation of this paper. All authors have read and agreed to the published version of the manuscript.

**Funding:** This research was funded by National Key R&D Program of China (2016YFE0131600), National Natural Science Foundation of China awarded project (41871371), A Project Funded by the Priority Academic Program Development of Jiangsu Higher Education Institutions, and by Ministry of Education, Youth and Sports of the Czech Republic (grant agreement no. LTACH-17002).

**Acknowledgments:** The authors acknowledge the Data Center of Yangtze River Delta, National Science & Technology Infrastructure to provide data in the use case study.

**Conflicts of Interest:** The authors declare no conflict of interest.

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
