# Peer review of "Constructing the CityGML ADE for the Multi-Source Data Integration of Urban Flooding"

_ijgi, doi:10.3390/ijgi9060359_

Round 1
Reviewer 1 Report
- General Comments:
This paper uses CityGML as the basis for developing an Application Domain Extension for waterlogging-related data. As stated by the authors: “A prototype system was implemented to integrate multi-source waterlogging-related data and provide data services for the waterlogging model and 3D dynamic visualization of waterlogging simulation results.”
The work seems promising.
However, the research motivation shown in the lines 48-55 is fragile. The authors say that:
-“In terms of urban waterlogging disaster simulation and analysis, there are a variety of urban waterlogging models that can more accurately simulate processes such as surface confluence, pipe network drainage, and land area water dispersion. However, these models often require a large amount of data as support, and these data show the characteristics of multi-source, heterogeneity, spatiotemporal differentiation, multi-resolution, multi-scale, etc., which brings great difficulty to data collection and application. Besides, most of the analysis results of these waterlogging models are presented in the form of text tables, which is not conducive to interpretation, transmission, and visualization in today's network environment”.
Urban flooding simulation is something done worldwide, using a variety of models (as said in the first sentence of the excerpt shown above). These models use a variety of data, which sometimes is difficult to obtain, as usually faced by deterministic models that represent natural phenomena, but this is not something that prevent modeling. Besides, the majority of the available models is capable of producing flood maps, flooding movies or graphs. I do agree with the authors when they say that “data integration and sharing for urban waterlogging simulation and analysis is particularly important”. Therefore, the authors could more carefully build their arguments around this last sentence.
Reading lines 126-128 makes me feel confused about what is going to be offered as a product. It is not clear to me, until this point, if the proposed module will be a complement to the hydrodynamic models previously cited in the text (line 105), in order to process and visualize results or if this module will be built with physical information regarding the system, pointing to zones with a higher propensity to flooding. These two options may be valid, but their uses are completely different. Serving as a post-processing tool may be very useful to materialize a flooding diagnosis, to support flood control design or disaster warning and communication of flood control alternatives. The deterministic hydrodynamic models can forecast future scenarios with accuracy and the combination of these tools with GML tools can be useful. On the other side, if a complete new model is being developed for estimating where are the critical points in terms of urban drainage behavior (depending on physical characteristics of the basin and the drainage network), its results are limited to a disaster planning phase. It is not possible to develop a project, for example, with an estimation of propensity to flooding. Another possibility would be to develop a complete system, integrating flood routing and CityGML. Therefore, I suggest the authors to make it clear what the contribution of the work is, from the beginning of the work. It is not positive to finish the first part of the paper without knowing exactly what its proposal is.
The manuscript Section 4 (“The city waterlogging ADE: data model”) showed a well structure proposal, with a very interesting approach to integrate flood routing and city 3D visualization, encompassing rainfall information, watershed behavior and drainage network representation. The application results seem to be useful. However, I would like to see/understand how water moves in the model. The authors use SWMM to simulate urban flooding, but SWMM usually gives information about surcharged manholes and overflow volumes. Overflows are not routed in SWMM. They are used to estimate the level of the failure of a node. How these volumes were routed over the streets in the CTWLADE? How water moves and how this movement is transformed into a visual 3D surface?
I am not an expert in GIS or CityGML and part of my difficulties can be derived from this fact. However, I am an expert in hydraulics and urban flood modeling. I think the proposed work is interesting, and I would be glad to have such a tool working, but there are grey points in the manuscript (at least, to me). This journal provides an advanced forum for the science and technology of geographic information, and, in this context, the proposed manuscript seems to bring an interesting approach to an important issue (urban flooding). However, acting as a researcher interested in urban flooding representation and visualization tools, I was not able to fully understand how the hydrodynamic modeling responses are used in the proposed module. The article, as a whole, would benefit from a clearer description of this interaction (as well as a wider audience would benefit too, including flood routing modelers). I would like to ask the authors to bring light to this interface between flood routing and the proposed module, to help in the clear understanding of the potentials of this work.
- Specific comments:
In the line 44, it seems that something is missing: “the construction of reasonable disaster emergency management” should possibly be “the construction of reasonable disaster emergency management plans” (or "strategies", or something else).
In the line 62, CityGML should be referred to a publication or a site or both.
In the line 105, references should be associated to the models SWMM, Mike Flood, InfoWorks ICM, Sewer GEMS.
Lines 118-120 must be rewritten. They are not understandable.
In the lines 250-251 there is a sentence not comprehensible: “…, which are represented by a self-aggregation method called consist of in the UML class diagram, …”
Figure 7 is difficult to read.
Author Response
Thank you for your comments concerning our manuscript entitled “Constructing the CityGML ADE for the multi-source data integration of urban flooding” (ID: ijgi-807360). Those comments are all valuable and very helpful for revising and improving our paper, as well as the important guiding significance to our researches. We have studied comments carefully and have made correction which we hope meet with approval.

Reviewer 2 Report
The topic covered in this article is of great importance for all those working in the modeling of flood phenomena. The high number of parameters involved in the modeling of the flood phenomena, as well as the complexity of the interaction between each parameter makes it difficult to take into account all the possible interactions which determine the accuracy of the results. For this reason, the models currently in use are generally simplified or have a different sensitivity to the parameters in use. And the main problem is undoubtedly the great sensitivity of any model to the geometric characteristics of the soil and the distribution of buildings and drainage systems.
So the methodological approach presented in this article is clearly of great use. However, the presentation of the method needs to be improved.
There are many technicalities in the presentation of the individual steps of the methodology which do not make reading easy.
In the section that describes the application of the method to a prototype, visual examples of the input files, both shape and excel, should be provided, so that it is easier to understand the interaction between the different modules explained in the previous sections.
Furthermore, the results presented in the application of the model in the prototype are not clear in the images: figure 13 for example does not provide relevant information. The difference between the different steps in terms of flood levels is unclear. A more detailed explanation of what is seen in the image is missing.
Can the results shown in the application prototype be validated in any way?
Author Response

(The authors gave the same response as above.)

Reviewer 3 Report
The paper presents a prototype system which can convert waterlogging-related multi-source data in XML files to the developed “City Waterlogging Application Domain Extension (CTWLADE)”. CTWLADE can map data from the hydraulic software SWMM and integrate it into a Web 3D Service to provide 3D dynamics visualization in interactive scenes. Authors present and explain the model modules, using some images to illustrate them; then, they test it in an application system using the Xianlin Campus of Nanjing Normal University in China as case study and present the obtained results. The main conclusions of the manuscript indicates that CTWLADE model can be useful to predict waterlogging, to estimate the risk of waterlogging disasters for old cities to reduce disaster losses, and to provide planning advice for new cities to prevent the occurrence of waterlogging.
The manuscript is in accordance with the aims and scope of the present journal.
Although the manuscript is well structured, I suggest that authors provide a native speaker review, to improve the written English in some points of the text.
In general terms, authors present an original work which provides an advance in current knowledge. However, some points should be improved.
I suggest that authors revise the abstract, rewriting the objective of the manuscript as it is not so clear in this part of the text. I also suggest that you explore better the results in this part of the text.
Sections 1 and 2 can be improved, with a deeper literature review, especially considering manuscripts from The International Journal of Geo-Information.
The main question to highlight is that the interaction with the hydrodynamic model is not clear. What is the role of the hydrodynamic model and what is the role of the presented framework?
Specific comments:
line 43-44: “...losses of 11.64 billion yuan.” Please, convert to dollars.
Please re-write lines 63-66 – very confusing sentence
Lines 71-76: authors mention ArcHydro and WaterML. Could you explore more and cite other models?
Lines 82-89 can be eliminated
Line 105 – please inform references for each hydrodynamic model mentioned
Line 118 English mistake “What are the last years has proven...”
line 125: please detail the application of CityGML in flood simulation mentioned by references 12 and 16.
lines 127-128: "Even CityGML falls partially short when it comes to the definition of specific entities and attributes fow waterlogging-related applications." Please detail this sentence explaining which are these entities.
Line 436 eliminate period after the word “Figure”
Figure 13: do authors have a more significant event to illustrate flood in this figures?
Regarding the references, there is only one from this journal. I suggest that authors try to find more related works in the proper International Journal of Geo-Information. This will reinforce the adherence of the presented research with this journal.
Author Response

(The authors gave the same response as above.)

Reviewer 4 Report
General comment
The paper in general is interesting and well structured. The study proposes a new data model that integrating multisource input and output data of waterlogging simulation and analysis and the demand for 3D visualization, and extending CityGML in waterlogging domain to form City Waterlogging Application Domain Extension. The focus of the study is that identifying a data model to support a more efficient disaster emergency management to reduce economic losses, social impacts and human casualties caused by natural disasters, which have become the top priority of sustainable urban development.
Specific comments for sections
- In the section 1. Introduction the authors introduce the general issue that they will address in the paper, setting the main objective of this study.
- In the section 2. Related work the authors propose a complete waterlogging model by integrating the simulation methods of different hydrological process. The most commonly used waterlogging model is the distributed hydrodynamic model, such as SWMM, Mike Flood, InfoWorks ICM, Sewer GEMS, they all manifest some criticalities in terms of data heterogeneity, computational and interpretative. To support an emergency management a 3D visualization is essential as well as a standardized and Omni-comprehensive urban data model covering the urban waterlogging domain.
- In the section 3. Analysis of urban waterlogging the authors describe the process of urban waterlogging in the sub-section 3.1, in which they highlight a schematic diagram of the basic flow urban production and confluence and they highlight the most significant elements for the design of the CTWLADE (sub-section 3.2) fairly clearly. They identify the urban waterlogging data source in three aspects: meteorological data, underlying surface data, and drainage pipe network data. They propose a synthesis the three macro areas data source in the Table 1 (sub-section 3.2.).
- In section 4. The city waterlogging ADE: Data Model the authors present in a clearly and consistent way all the steps which lead to the definition of the new data model proposed that satisfied all their requirement as integrating multisource input and output data of waterlogging simulation and analysis and the demand for 3D visualization, and extending CityGML in waterlogging domain to form City Waterlogging Application Domain Extension (sub-section 4.1 4.2, 4.3:4.3.1.,4.3.2, 4.4, 4.5, 4.6, 4.7).
- In the section 5. Data integration and Dynamic 3D visualization the authors verified the validity the CTWLADE by a prototype system. At end of this section 5 the authors highlight some strengths of the proposed model: the data integration and share of waterlogging data based on the CTWLADE model can be developed independently without worrying about whether the data source format needs to be compatible with the data requirements of the waterlogging simulation thus reducing the efforts on how to integrate and organize the multi-source data; provide a data-sharing service that is easy to render and contains attribute information for dynamic 3D visualization thus expanding the readability and practicability of the simulation results of waterlogging; provide very intuitive and clear data support for disaster management, thus aiding to disaster warning and decision-making.
- In the section 6. Conclusion the authors proposed a brief summary of the items studied in this paper and highlighted some possible development directions and improvements of the model proposed by them.
Author Response

(The authors gave the same response as above.)

Round 2
Reviewer 2 Report
I consider that after the changes made by the authors, the article in this version, is ready to be published.
Best regards